# Distribution Guidance Network for Weakly Supervised Point Cloud Semantic Segmentation

**Zhiyi Pan**
SECE, Peking University
Peng Cheng Laboratory
panzhiyi@stu.pku.edu.cn

**Wei Gao**
SECE, Peking University
gaowei262@pku.edu.cn

**Shan Liu**
Media Laboratory, Tencent
shanl@tencent.com

**Ge Li**[*]
SECE, Peking University
geli@ece.pku.edu.cn

## Abstract

Despite alleviating the dependence on dense annotations inherent to fully supervised methods, weakly supervised point cloud semantic segmentation suffers from inadequate supervision signals. In response to this challenge, we introduce a novel perspective that imparts auxiliary constraints by regulating the feature space under weak supervision. Our initial investigation identifies which distributions accurately characterize the feature space, subsequently leveraging this priori to guide the alignment of the weakly supervised embeddings. Specifically, we analyze the superiority of the mixture of von Mises-Fisher distributions (moVMF) among several common distribution candidates. Accordingly, we develop a **D**istribution **G**uidance **Net**work (DGNet), which comprises a weakly supervised learning branch and a distribution alignment branch. Leveraging reliable clustering initialization derived from the weakly supervised learning branch, the distribution alignment branch alternately updates the parameters of the moVMF and the network, ensuring alignment with the moVMF-defined latent space. Extensive experiments validate the rationality and effectiveness of our distribution choice and network design. Consequently, DGNet achieves state-of-the-art performance under multiple datasets and various weakly supervised settings.

## 1 Introduction

As a fundamental task in 3D scene understanding, point cloud semantic segmentation [42, 32, 29] is widely entrenched in 3D applications, such as 3D reconstruction [15, 33], autonomous driving [20], and embodied intelligence [14, 50]. Despite significant accomplishments in tackling the disorder and disorganization, point cloud semantic segmentation remains annotation-intensive, hindering its expansion in big datasets and large models. For this reason, the academic community explores achieving point cloud semantic segmentation in a weakly supervised manner. However, due to the lack of supervision signals, learning point cloud segmentation on sparse annotations is nontrivial.

In recent years, considerable effort has been made to pursue additional constraints in weak supervision. As shown in Fig. 1, representative work can be broadly categorized into several paradigms: 1) *Contrastive Learning / Perturbation Consistency* imposes contrastive loss or consistency constraint between network embeddings of original and perturbed point clouds, respectively. 2) *Self-training* progressively enhances segmentation quality by treating reliable predictions as pseudo-labels, with

---

[*]Ge Li is the corresponding author.

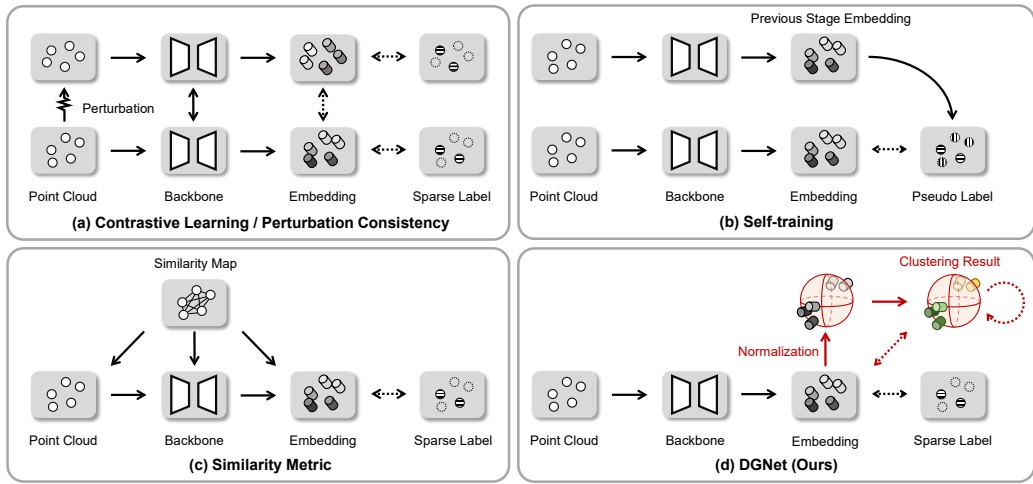

Figure 1: Visual comparisons of mainstream weakly supervised point cloud semantic segmentation paradigms and our DGNet. The solid and dashed lines represent the network forward process and the loss function, respectively.

given sparse annotations as initial labels. 3) *Similarity Metric* transfers supervision signals from annotated points to unlabeled regions via leveraging the low-level features or the embedding similarity. Nevertheless, most constraints stem from heuristic assumptions and ignore the inherent distribution of network embedding, resulting in ambiguous interpretations of point-level predictions. In contrast to existing paradigms, in this paper, we re-examine two fundamental issues: *How to characterize the semantic segmentation feature space for weak supervision and how to intensify this intrinsic distribution under weakly supervised learning?*

For the first issue about `oughtness`, we expect to provide a mathematically describable distribution for weakly supervised features. Consequently, we attempt to precisely describe the feature space in terms of two dimensions: distance metric and distribution modeling. In distance metric, we compare the Euclidean norm and cosine similarity between representations, and in distribution modeling, the category prototype model and the mixture model are considered. Among our candidate combinations, the mixture of von Mises-Fisher distributions (moVMF) with cosine similarity is finalized, due to its powerful fitting capability to segment head and insensitivity to the Curse of Dimensionality [44]. We believe that a superior weakly supervised feature space should adhere to this distribution.

For the second issue about `practice`, we dynamically align the embedding distribution in the hidden space to moVMF during weakly supervised learning. Accordingly, we propose a **D**istribution **G**uidance **Net**work (DGNet), comprising a weakly supervised learning branch and a distribution alignment branch. Specifically, the weakly supervised learning branch learns semantic embeddings under sparse annotations, while the distribution alignment branch constrains the distribution of the network embeddings. Via a Nested Expectation-Maximum Algorithm, the semantic features are dynamically refined. Therefore, restricting and fitting is a mutually reinforcing, iterative optimization process. To curtail the pattern of feature distribution, we derive the vMF loss based on the maximum likelihood estimation and the discriminative loss inspired by metric learning [22]. For joint optimization, consistency loss is imposed between the segmentation predictions and the posterior probabilities. During the inference phase, only the weakly supervised learning branch is activated to maintain inference consistency with fully supervised learning.

We validate DGNet on three prevailing point cloud datasets, *i.e.*, S3DIS [1], ScanNetV2 [11], and SemanticKITTI [5]. After the constraints of feature distribution, DGNet provides significant performance improvements over multiple baselines. Across various label rates, our method achieves state-of-the-art weakly supervised semantic segmentation performance. Extensive ablation studies also confirm the effectiveness of each loss term we proposed. In addition, posterior probabilities under the moVMF provide a plausible interpretation for predictions on unlabeled points.

## 2  Related Work

**Weakly Supervised Point Cloud Semantic Segmentation.**   Weakly supervised point cloud semantic segmentation methods aim to provide reliable additional supervision with sparse annotations. Four paradigms have been successively proposed in recent years, *i.e.*, perturbation consistency, contrastive learning, self-training, and similarity metric. Perturbation consistency methods are based on the assumption of perturbation invariance of network features, imposing diverse perturbations (such affine transforms with point jitter [59, 54], downsampling [56], masking [35], etc.) to construct pairs of point clouds. Several methods [31, 34] introduce contrastive learning in weak supervision to encourage the discriminability of hidden layer features. Additionally, pre-training methods [55, 17] with contrastive learning similarly demonstrate the ability to bias induction in the face of downstream semantic segmentation tasks with sparse annotations. Self-training methods generate reliable dynamic pseudo-labels based on previous stage predictions for subsequent training stages. Via CAMs [62], MPRM [51] and J2D3D [25] dynamically generate point-wise pseudo-labels from subcloud-level annotations and image-level annotations, respectively. Recently, REAL [24] integrate SAM [23] to self-training. Similarity metric methods measure the similarity between labeled and unlabeled points to propagate supervision information, in which the similarity is elaborated on low-level features [49, 52], network embedding [36, 18, 40] or category prototypes [58, 46]. Most similar to our method is the similarity metric strategy. However, the distinction is that DGNet focuses on describing network embeddings holistically, rather than constructing pair relationships between features.

**Feature Distribution Constraints.**   The constraints of feature distribution are always presented in the form of feature clustering. DeepClustering [6], which integrates clustering and unsupervised feature learning, utilizes the clustering results as pseudo-labels to extract visual features dynamically. Following this groundbreaking work, a series of subsequent studies [7, 2, 30] apply feature clustering in unsupervised learning to obtain discriminative visual features for pretraining. For point cloud semantic segmentation, PointDC [10] delineates semantic objects by aligning the features on the same super-voxel in an unsupervised manner. Feng *et al*. [12] imposes a clustering-based representation learning to enhance the discrimination of embeddings under full supervision. In contrast, our DGNet is oriented towards weakly supervised semantic segmentation by restricting the feature distribution.

**Mixture of von Mises-Fisher Distributions.**   The moVMF [3] describes the embeddings of multiple categories on the unit hypersphere in feature space, where the parameters are jointly optimized with the clustering results by Expectation-Maximum algorithm [3, 4]. Some work attempts to combine neural networks with the moVMF in the deep learning era. For example, [16] view face verification as a direct application of clustering, introducing a vMF loss to align the distribution of face features. Segsort [21], on the other hand, utilizes the prior of the moVMF to over-segment images. DINO-VMF [13] achieve a more stable pre-trained method by precisely describing DINO [8] as a moVMF. In this work, we discuss the superiority of moVMF in characterizing semantic embeddings and trust it as a priori to guide weakly supervised learning.

## 3  Methodology

### 3.1  Preliminaries

**Task Definition.**   Without loss of generality, a point cloud for weakly supervised learning is denoted as $\{(\mathbf{X}_l, \mathbf{Y}), (\mathbf{X}_u, \varnothing)\} = \{(\mathbf{x}_1, y_1), \cdots, (\mathbf{x}_m, y_m), \mathbf{x}_{m+1}, \cdots, \mathbf{x}_n\}$, where $\mathbf{X}_l$ and $\mathbf{X}_u$ are the point sets with and without annotations, respectively. $\mathbf{Y}$ is the corresponding annotations on $\mathbf{X}_l$, in which $y_i \in \mathbb{C}$ and $\mathbb{C}$ is the set of category indices. $n$ and $m$ are the point numbers of the point cloud and labeled set, respectively. Fed into the segment head, the network embedding $\mathbf{f}_i$ is projected into the category probability vector $\mathbf{p}_i$. The partial cross-entropy loss is employed in conventional weakly supervised semantic segmentation:

$$\mathcal{L}_{\mathrm{pCE}} = -\frac{1}{m} \sum_{i=1}^{m} \log(\mathbf{p}_i^{y_i}), \tag{1}$$

where $\mathbf{p}_i^{y_i}$ represents the probability of $y_i$-th category in $\mathbf{p}_i$.

**von Mises-Fisher Distribution (vMF).** The vMF has demonstrated strong data fitting and generalization capabilities in the fields of self-supervised learning [9, 13], classification [44], variational inference [47], and online continual learning [38]. This distribution describes the distribution of normalized embedding $\mathbf{v}_i = \text{norm}(\mathbf{f}_i)$ on the unit hypersphere, with the probability density function:

$$f(\mathbf{v}_i|\mathbf{u}, \kappa) = C_d(\kappa)\exp(\kappa\mathbf{u}^\top\mathbf{v}_i), \tag{2}$$

where $\mathbf{u}$ represents the mean vector of vMF and $\kappa \geq 0$ is a concentration parameter that controls the probability concentration around $\mu$. $C_d(\kappa)$ is the normalization constant.

**Mixture of vMF (moVMF).** Similar to other mixture models, moVMF treats vMF as a sub-distribution to describe the overall distribution of multiple categories. Over the entire set of categories $\mathbb{C}$, the probability density function of moVMF is formulated as:

$$P(\mathbf{v}_i|\mathbb{C}, \Theta) = \sum_{c\in\mathbb{C}}\alpha_c f(\mathbf{v}_i|\mathbf{u}_c, \kappa_c) = \sum_{c\in\mathbb{C}}\alpha_c C_d(\kappa_c)\exp(\kappa_c\mathbf{u}_c^\top\mathbf{v}_i), \tag{3}$$

where $\Theta = \{\alpha_c, \kappa_c, \mathbf{u}_c | c \in \mathbb{C}\}$ is the parameters of moVMF. $\alpha_c$ denotes the proportion of the von Mises-Fisher distribution for the $c$-th category and $\sum\alpha_c = 1$.

## 3.2 Feature Space Description

We intend to provide additional supervision signals for weakly supervised learning by portraying and enhancing its inherent distribution. Specifically, We explore it in two dimensions, *i.e.*, the distance metric and the distribution modeling:

- **Distance metric.** Distance metric influences the similarity relationship between features. We consider the two most commonly used distance measures in clustering, *i.e.*, Euclidean norm and cosine similarity. For given vectors $\mathbf{u}$ and $\mathbf{v}$, the Euclidean norm is defined as $\|\mathbf{u} - \mathbf{v}\|_2$ and the cosine similarity is defined as $\frac{\mathbf{u}^\top\mathbf{v}}{\|\mathbf{u}\|_2\|\mathbf{v}\|_2}$. Cosine similarity can be viewed as the inner product of normalized $\mathbf{u}$ and $\mathbf{v}$.

- **Distribution modeling.** Distribution modeling determines the clustering results of features. A straightforward model is the Category Prototype [45]. In this model, clusters are assigned by comparing the distance between the features and each category prototype. In addition, we incorporate mixture models into the comparison. Depending on the distance measure, we categorize it into the Gaussian Mixture Model (GMM) with Euclidean norm and the mixture of von Mises-Fisher distributions (moVMF) with cosine similarity, respectively.

Describing the feature space of a neural network remains an open problem. Various factors influence the feature space, including network architecture, training data, parameter configurations, and the optimization (loss) function. Given this intractability, deriving a universally optimal mathematical description is impractical. Therefore, we discuss the merits and demerits of these candidate distributions from the following perspectives:

- **Segment head.** To facilitate the analysis, we simplify the structure of the segment head as $\text{SegHead}(\mathbf{f}) = \text{argmax}(\text{softmax}(\mathbf{w}\mathbf{f}^\top))$, where $\mathbf{f}$ is the semantic feature extracted by the decoder and $\mathbf{w}$ is the parameter of the output layer. Consider a group of feature vectors $\{k\mathbf{f}|k \geq 0 \ \& \ \mathbf{f} \neq \mathbf{0}\}$. For any two feature vectors $k_1\mathbf{f}$ and $k_2\mathbf{f}$ within this group, the segmentation predictions are identical, *i.e.*, $\text{SegHead}(k_1\mathbf{f}) = \text{argmax}(\text{softmax}(k_1\mathbf{w}\mathbf{f}^\top)) = \text{argmax}(\text{softmax}(k_2\mathbf{w}\mathbf{f}^\top)) = \text{SegHead}(k_2\mathbf{f})$. If the general case of using an activation function is taken into account, it does not change the result after argmax since the activation function is usually monotonically nondecreasing. Therefore, the segment head is a radial classifier with a more pronounced classification performance on the angles, so cosine similarity describes the feature space better than the Euclidean norm.

- **Curse of dimensionality.** Another advantage of cosine similarity can be explained in terms of the Curse of Dimensionality. Most high-dimensional features are far from each other, causing the Euclidean distance to become ineffective in distinguishing differences between feature vectors. Cosine similarity, on the other hand, is more effective in distinguishing differences between features by measuring the angle between the vectors. Besides, the Euclidean norm is sensitive to scale while cosine similarity is not affected by the length of the vectors.

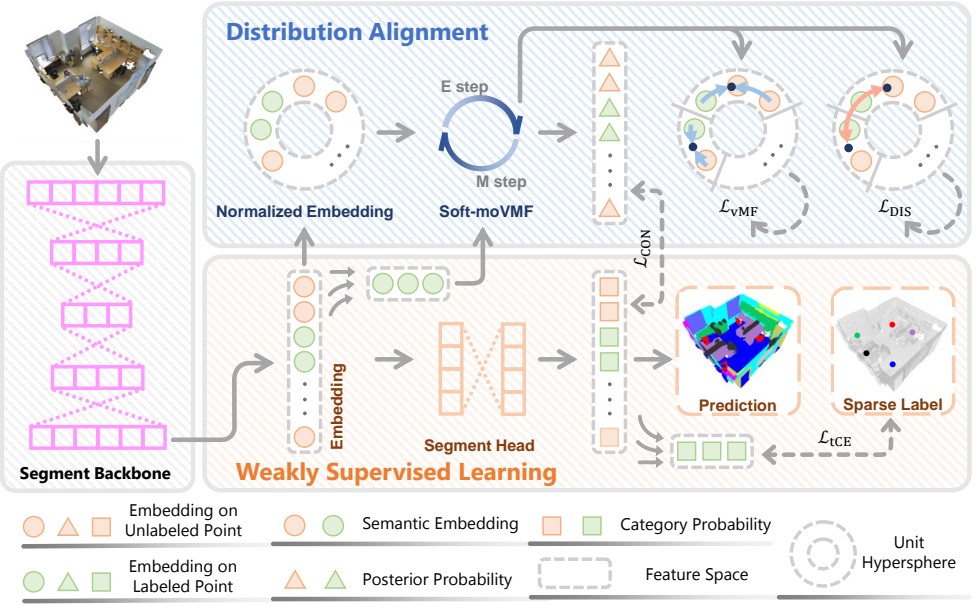

Figure 2: Structure of Distribution Guidance Network.

- **Fitting ability.** Despite its computational simplicity, the Category Prototype Model clusters the features by comparing the distance between features and category prototypes. This means that the Category Prototype Model ignores the distribution within categories and the variability between categories. In contrast, the Mixture Model possesses intra-category fitting and inter-category perception capabilities.

Based on the above analysis and experimental validation in Sec. 4.3, we characterize the feature space as moVMF and propose the Distribution Guidance Network to enhance this distribution.

### 3.3 Distribution Guidance Network

To enhance the intrinsic distributions discussed in Section 3.2, we propose a **D**istribution **G**uidance **Net**work (DGNet). The structure of DGNet is shown in Fig. 2, which comprises the weakly supervised learning branch and the distribution alignment branch.

#### 3.3.1 Weakly Supervised Learning Branch

Sparse annotations pose two challenges for the point cloud semantic segmentation. The first is underfitting the entire dataset, as the supervision signals are insufficient for complex structured point clouds. To address underfitting, we introduce additional signals by reinforcing the inherent distribution of the feature space. The second challenge is overfitting within the labeled set $\mathbf{X}_l$, as the model capacity is more than adequate to fit the labeled points. To mitigate overfitting, we replace the conventional partial cross-entropy loss Eq. 1 with the truncated cross-entropy loss [60] in the weakly supervised learning branch, which is defined as:

$$\mathcal{L}_{\text{tCE}} = -\frac{1}{m} \sum_{i=1}^{m} \min \Big( \log(\mathbf{p}_i^{y_i}), \log(\beta) \Big), \tag{4}$$

where $\beta \in [0, 1]$ represents the threshold for truncating the cross-entropy loss. With $\mathcal{L}_{\text{tCE}}$, the gradient of the cross-entropy loss is curtailed when the predicted class probability for an annotated point exceeds $\beta$. Over-optimization for that point is halted, thereby preventing overfitting.

The weakly supervised learning branch also provides robust initialization for the distribution alignment branch by utilizing the average feature vector of the labeled points. According to the Central Limit Theorem [27], the difference between the initialization vector from the weakly supervised

learning branch and the theoretical optimal average vector conforms to a Gaussian distribution with a mean of 0. The initialized mean vector in DGNet has a high probability of appearing in the vicinity of the optimal solution, which facilitates the clustering algorithm in achieving rapid and stable convergence.

### 3.3.2 Distribution Alignment Branch

The feature space descriptor moVMF, investigated in Sec. 3.2, is employed to regulate the feature space under weakly supervised learning. Initially, the network embeddings are normalized and projected onto the unit hyperspherical surface of the feature space, *i.e.*, $\mathbf{v}_i = \text{norm}(\mathbf{f}_i)$. Following this, the optimization objective function is defined utilizing maximum likelihood estimation[1]:

$$\max_{\phi, \mathcal{Z}, \Theta} P(\mathcal{V}|\mathcal{Z}, \Theta) = \min_{\phi, \mathcal{Z}, \Theta} -\sum_{i=1}^{n} \Big[ \log(\alpha_{z_i}) + \kappa \mathbf{u}_{z_i}^{\top} \mathbf{v}_i \Big], \tag{5}$$

where $\phi$, $\mathcal{V} = \{\mathbf{v}_i\}$, $\mathcal{Z} = \{z_i\}$ and $\Theta = \{\alpha_c, \kappa, \mathbf{u}_c\}$ denote the learnable network parameters, the normalized network embeddings, the corresponding clustering results and the parameters of moVMF, respectively. To avoid the long-tail problem and simplify computations, the concentration parameter $\kappa$ is fixed as a constant in our implementation. Since the clustering initialization is category-aware, the clustering results $z_i \in \mathbb{C}$ are with category labels. While the primary optimization goal is the learning of network parameters $\phi$, we dynamically resolve $\mathcal{Z}$ and $\Theta$ to furnish a more precise feature space description. Consequently, we develop a Nested Expectation-Maximum Algorithm to manage the challenge associated with the three optimization variables delineated in Eq. 5.

- **E Step (Optimize $\Theta$ and $\mathcal{Z}$):** Regarding network embeddings as input conditions, we integrate the soft-moVMF EM algorithm [3] into the network to alternately optimize $\Theta$ and $\mathcal{Z}$. To enhance the stability and computational efficiency of the algorithm, we utilize the average features of labeled points from the weakly supervised branch to initialize $\mathbf{u}$. The posterior probability set $\mathcal{Q} = \{\mathbf{q}_i\}$ serves as the soft assignment for the clustering results $\mathcal{Z}$, where $\mathbf{q}_i$ is defined as

$$\mathbf{q}_i = P(c|\mathbf{v}_i, \Theta) = \frac{\alpha_c \exp(\kappa \mathbf{u}_c^{\top} \mathbf{v}_i)}{\sum_{l \in \mathbb{C}} \alpha_l \exp(\kappa \mathbf{u}_l^{\top} \mathbf{v}_i)}. \tag{6}$$

Compared to other algorithms in [3], the soft-moVMF algorithm updates $\Theta$ by weighting all features according to their posterior probabilities, considering inter-cluster similarities, thereby achieving more accurate parameter updates. The posterior probability $\mathbf{q}_i \in [0,1]^{|\mathbb{C}| \times 1}$ is employed not only as a weighting factor for updates but also in the calculation of the loss function for joint optimization. Furthermore, $\mathcal{Q}$ provides a probabilistic explanation for the predictions during the inference phase.

The complexity of the soft-voVMF is $O(tn|\mathbb{C}|)$, where $t$ is the iteration number, $n$ is the point number of the point cloud, and $|\mathbb{C}|$ is the number of semantic categories. Since $t$, $n$, and $|\mathbb{C}|$ are all set to constant values during network training, the extra computation introduced by the distribution alignment branch is trivial.

- **M Step (Optimize $\phi$):** With the converged parameters $\Theta$ and $\mathcal{Z}$ fixed, we optimize $\phi$ by the backpropagation process. Consistent with the philosophy of the soft-moVMF, we incorporate the posterior probability $\mathcal{Q}$ into Eq. 5, and reformulate it into the loss function as follows:

$$\mathcal{L}_{\text{vMF}} = -\sum_{i=1}^{n} \sum_{c \in \mathbb{C}} \mathbf{q}_i^c \Big[ \log(\alpha_c) + \kappa \mathbf{u}_c^{\top} \mathbf{v}_i \Big]. \tag{7}$$

Additionally, acknowledging the significance of distinct decision boundaries within the mixture model, we incorporate a discriminative loss derived from metric learning [22] which is defined as:

$$\mathcal{L}_{\text{DIS}} = \frac{1}{|\mathbb{C}|(|\mathbb{C}| - 1)} \sum_{c_1, c_2 \in \mathbb{C} \& c_1 \neq c_2} \mathbf{u}_{c_1}^{\top} \mathbf{u}_{c_2}. \tag{8}$$

The interpretation of Bayesian posterior probabilities for predictions based on the moVMF is an attractive property of DGNet. Fig. 3 visualizes the posterior probabilities for some categories. Taking the `floor` as an example, according to the Bayesian theorem, those points with relatively high posterior probabilities are more likely to be `floor`, which explains the prediction results.

---

[1]A complete reasoning process can be found in the Appendix.

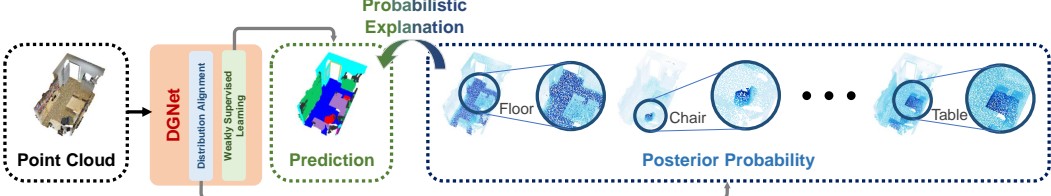

Figure 3: DGNet provides segmentation predictions from the weakly supervised learning branch and explains it probabilistically by posterior probabilities from the distribution alignment branch.

### 3.3.3 Loss Function

In addition to the previously mentioned initialization, we introduce a consistency loss to fortify the exchange of information between the two branches. This consistency loss is imposed on the class probability map $\mathbf{p}$ from the weakly supervised learning branch and the posterior probability $\mathbf{q}$ from the distribution alignment branch, in the form of cross-entropy:

$$\mathcal{L}_{\mathrm{CON}} = -\frac{1}{n}\sum_{i=1}^{n}\mathbf{q}_i^{\top}\log(\mathbf{p}_i). \tag{9}$$

If regard the posterior probability $\mathbf{q}$ as pseudo-labels, the consistency loss is proved to diminish prediction uncertainty and alleviate distribution discrepancies in [48].

Without laborious adjustments to the weights[2], the overall loss function is defined as follows:

$$\mathcal{L} = \mathcal{L}_{\mathrm{tCE}} + \mathcal{L}_{\mathrm{vMF}} + \mathcal{L}_{\mathrm{DIS}} + \mathcal{L}_{\mathrm{CON}}. \tag{10}$$

## 4 Experimental Analysis

### 4.1 Experiment Settings

**Datasets.** S3DIS [1] encompasses six indoor areas, constituting a total of 271 rooms with 13 categories. Area 5 within S3DIS serves as the validation set, while the remaining areas are allocated for network training. ScanNetV2 [11] offers a substantial collection of 1,513 scanned scenes originating from 707 indoor environments with 21 indoor categories. Adhering to the official ScanNetV2 partition, we utilize 1,201 scenes for training and 312 scenes for validation. SemanticKITTI [5] with 19 classes is also considered. Point cloud sequences 00 to 10 are used in training, with sequence 08 as the validation set. To simulate sparse annotations, we randomly discard the dense annotations proportionally.

**Implementation details.** ResGCN-28 in DeepGCN [29] and PointNeXt-l [43] are reimplemented as the segment backbones with OpenPoints library [43]. We discard the last activation layer of the decoder to extract orientation-completed feature space. We maintain a memory bank [53] to store class prototypes across the entire dataset. In cases where class annotations are absent from the scene, the class prototypes from the memory bank are employed as supplementary initialization. We employ the LaDS [39] to maintain a higher rate of training supervision after point cloud sampling. For truncated cross-entropy loss, $\beta = 0.8$. The concentration constant $\kappa = 10$ and the iteration number $t = 10$. The distribution alignment branch is not activated in the first 50 epochs to stabilize the feature learning. In our implementation, the DGNet is trained with one NVIDIA V100 GPU on S3DIS, eight NVIDIA TESLA T4 GPUs on ScanNetV2, and one NVIDIA V100 GPU on SemanticKITTI. In the inference stage, only the weakly supervised learning branch is activated to produce predictions.

### 4.2 Comparative Analysis

**Results on S3DIS.** We detail the segmentation performance at 0.1% and 0.01% label rates on S3DIS Area 5. DGNet boosts performance for each baseline, which is evenly distributed across categories.

---

[2]The experiments demonstrate that DGNet is not sensitive to loss term weights.

Table 1: Quantitative comparisons on S3DIS Area 5 under various weakly supervised settings. The **bold** denotes the best performance.

| Setting | Method | mIoU | ceiling | floor | wall | beam | column | window | door | chair | table | bookcase | sofa | board | clutter |
|---|---|---|---|---|---|---|---|---|---|---|---|---|---|---|---|
| 100% (Fully) | PointNet [41] | 41.1 | 88.8 | 97.3 | 69.8 | **0.1** | 3.9 | 46.3 | 10.8 | 59.0 | 52.6 | 5.9 | 40.3 | 26.4 | 33.2 |
| | SQN [18] | 63.7 | 92.8 | 96.9 | 81.8 | 0.0 | 25.9 | 50.5 | 65.9 | 79.5 | 85.3 | 55.7 | 72.5 | 65.8 | 55.9 |
| | HybridCR [31] | 65.8 | 93.6 | 98.1 | 82.3 | 0.0 | 24.4 | 59.5 | 66.9 | 79.6 | 87.9 | 67.1 | 73.0 | 66.8 | 55.7 |
| | ERDA [48] | 68.3 | 93.9 | **98.5** | 83.4 | 0.0 | 28.9 | 62.6 | 70.0 | **89.4** | 82.7 | 75.5 | 69.5 | 75.3 | 58.7 |
| | DeepGCN [29] | 60.0 | 90.8 | 97.5 | 76.7 | 0.0 | 24.0 | 51.4 | 52.7 | 76.7 | 83.0 | 61.1 | 62.2 | 58.5 | 44.6 |
| | PointNeXt [43] | 69.2 | **94.7** | 98.5 | 82.9 | 0.0 | 24.2 | 59.9 | **74.3** | 83.0 | **91.4** | **76.3** | 75.5 | **78.6** | **60.4** |
| | PointTransV1 [61] | 70.4 | 94.0 | 98.5 | 86.3 | 0.0 | **38.0** | 63.4 | 74.3 | 89.1 | 82.4 | 74.3 | **80.2** | 76.0 | 59.3 |
| 0.1% | SQN [18] | 64.1 | 91.7 | 95.6 | 78.7 | 0.0 | 24.2 | 55.9 | 63.1 | 70.5 | 83.1 | 60.7 | 67.8 | 56.1 | 50.6 |
| | CPCM [35] | 66.3 | 91.4 | 95.5 | **82.0** | 0.0 | **30.8** | 54.1 | 70.1 | 79.4 | 87.6 | 67.0 | 70.0 | **77.8** | 56.6 |
| | PointMatch [52] | 63.4 | - | - | - | - | - | - | - | - | - | - | - | - | - |
| | AADNet [39] | 67.2 | 93.7 | 98.0 | 81.5 | 0.0 | 19.4 | **59.5** | **72.0** | 80.9 | 88.5 | **78.3** | 73.0 | 72.1 | 56.1 |
| | DeepGCN [29] | 43.9 | 93.4 | 97.6 | 68.3 | 0.0 | 19.6 | 39.6 | 4.9 | 47.4 | 35.2 | 59.3 | 50.2 | 2.2 | 32.2 |
| | + DGNet | 58.4 | 91.2 | 97.3 | 76.5 | 0.0 | 22.7 | 47.2 | 42.8 | 73.2 | 85.0 | 62.0 | 59.7 | 58.1 | 44.2 |
| | PointNeXt [43] | 65.0 | 93.7 | 97.8 | 79.5 | 0.0 | 27.3 | 59.2 | 62.2 | 79.4 | 88.6 | 64.2 | 70.1 | 69.3 | 53.6 |
| | + DGNet | 67.8 | 94.3 | 98.4 | 81.6 | 0.0 | 28.9 | 57.2 | 70.5 | 82.3 | 90.7 | 74.1 | 75.2 | 70.3 | 58.5 |
| 0.03% | PSD [59] | 48.2 | 87.9 | 96.0 | 62.1 | 0.0 | 20.6 | 49.3 | 40.9 | 55.1 | 61.9 | 43.9 | 50.7 | 27.3 | 31.1 |
| 0.03% | HybridCR [31] | 51.5 | 85.4 | 91.9 | 65.9 | 0.0 | 18.0 | 51.4 | 34.2 | 63.8 | 78.3 | 52.4 | 59.6 | 29.9 | 39.0 |
| 0.03% | DCL [57] | 59.6 | 91.7 | 95.8 | 76.4 | 0.0 | 21.2 | 58.3 | 29.6 | 72.6 | 83.3 | **64.2** | 69.6 | 63.4 | 48.6 |
| 0.02% | MILTrans [56] | 51.4 | 86.6 | 93.2 | 75.0 | 0.0 | 29.3 | 45.3 | 46.7 | 60.5 | 62.3 | 56.5 | 47.5 | 33.7 | 32.2 |
| 0.02% | ERDA [48] | 48.4 | 87.3 | 96.3 | 61.9 | 0.0 | 11.3 | 45.9 | 31.7 | 73.1 | 65.1 | 57.8 | 26.1 | 36.0 | 36.4 |
| 0.02% | MulPro [46] | 47.5 | 90.1 | 96.3 | 71.8 | 0.0 | 6.7 | 46.7 | 39.2 | 67.2 | 67.4 | 21.8 | 39.2 | 33.0 | 38.0 |
| 0.01% | SQN [18] | 45.3 | 89.2 | 93.5 | 71.3 | 0.0 | 4.1 | 34.7 | 41.0 | 54.9 | 66.9 | 25.7 | 55.4 | 12.8 | 39.6 |
| 0.01% | CPCM [35] | 59.3 | - | - | - | - | - | - | - | - | - | - | - | - | - |
| 0.01% | AADNet [39] | 60.8 | 92.5 | 96.6 | 77.2 | 0.0 | 20.9 | 57.0 | 61.1 | 72.2 | 83.1 | **60.1** | 67.8 | 52.9 | 49.0 |
| 0.01% | DeepGCN [29] | 35.9 | 76.7 | 97.1 | 65.2 | 0.0 | 0.0 | 0.4 | 8.1 | 57.6 | 58.0 | 14.9 | 49.1 | 8.5 | 28.2 |
| 0.01% | + DGNet | 52.8 | 92.0 | 97.9 | 75.2 | 0.0 | 23.4 | 22.7 | 33.4 | 74.1 | 83.6 | 29.8 | 62.0 | 47.1 | 45.1 |
| 0.01% | PointNeXt [43] | 58.4 | 89.4 | 96.5 | 75.7 | **0.1** | 22.6 | **55.3** | 44.5 | 74.2 | 84.3 | 54.2 | 62.8 | 52.5 | 47.0 |
| 0.01% | + DGNet | 62.4 | 93.3 | 98.1 | 80.1 | 0.0 | **23.3** | 47.9 | 53.1 | 79.4 | 87.2 | 60.0 | 70.6 | 65.2 | 53.1 |

The lower the label rate, the more distinct the enhancement brought by DGNet, which suggests that the guidance on feature distribution is more valuable with extremely sparse annotations. Specifically, DGNet achieves more than 97% performance of fully-supervision with only 0.1% points labeled. The 0.02% label rate denotes a sparse labeling form of "one-thing-one-click". DGNet outperforms these methods without introducing super-voxel information. In addition, Fig. 4 visualizes a qualitative comparison. DGNet provides a holistic enhancement to the baseline. Although the photos on the wall are misclassified, DGNet captures consistent objects more accurately than the baseline.

**Results on ScanNetV2.** Compared to S3DIS, ScanNetV2 involves diverse categories and versatile scenes. Therefore, following the super-voxel setting in OTOC [36], we report the segmentation performances with 20 labeled points per scene and 1% points labeled. The cross-entropy loss term generates relatively sufficient supervised information to train the network due to introducing pseudo-labeling, resulting in a less pronounced DGNet improvement than S3DIS. However, DGNet is still slightly superior to the latest SOTA methods.

**Results on SemanticKITTI.** DGNet performs excellently on indoor datasets and demonstrates strong weakly supervised learning efficiency on outdoor SemanticKITTI. For a fair comparison, we replace the DGNet backbone with RandLA-Net [19]. DGNet outperforms SQN [18] with 1.0% and 2.1% mIoU on 0.1% and 0.01% label rates, respectively, demonstrating the necessity of supervision on feature space.

Table 2: Quantitative comparisons on ScanNet.

| Setting | Method | mIoU (%) |
|---|---|---|
| 100% (Fully) | PointNet [41] | 33.9 |
| | HybirdCR [31] | 59.9 |
| | PointNeXt [43] | 71.2 |
| 1% | Zhang et al. [58] | 51.1 |
| | PSD [59] | 54.7 |
| | HybirdCR [31] | 56.8 |
| | DCL [57] | 59.6 |
| | GaIA [28] | 65.2 |
| | EDRA [48] | 63.0 |
| | AADNet [39] | 66.8 |
| | DGNet (PointNeXt) | **67.4** |
| 20pts | Hou et al. [17] | 55.5 |
| | OTOC [36] | 59.4 |
| | MILTrans [56] | 54.4 |
| | DAT [54] | 55.2 |
| | PointMatch [52] | 62.4 |
| | EDRA [48] | 57.0 |
| | AADNet [39] | 62.5 |
| | DGNet (PointNeXt) | **62.9** |

Table 3: Quantitative comparisons on SemanticKITTI.

| Setting | Method | mIoU (%) |
|---|---|---|
| 0.1% | RPSC [26] | 50.9 |
| | SQN [18] | 50.8 |
| | DGNet (RandLA-Net) | **51.8** |
| 0.01% | CPCM [35] | 34.7 |
| | SQN [18] | 39.1 |
| | DGNet (RandLA-Net) | **41.2** |

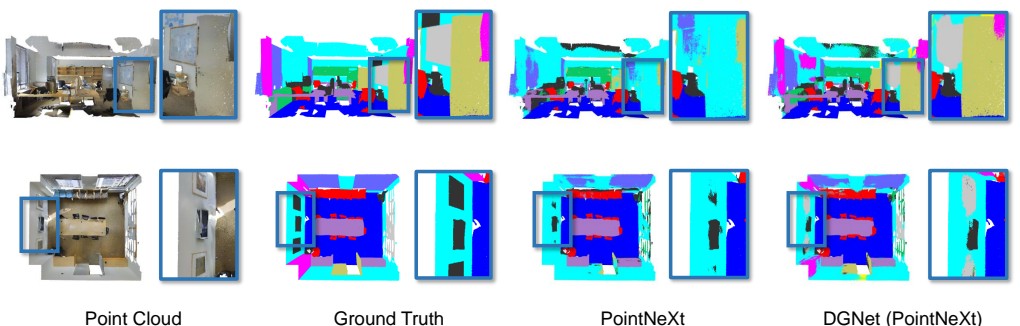

| Point Cloud | Ground Truth | PointNeXt | DGNet (PointNeXt) |

Figure 4: Visual comparisons between baseline and our DGNet on S3DIS Area 5 at 0.01% label rate.

Table 4: Comparisons on feature distribution description selection in distribution alignment branch.

| Distribution | Distribution Modeling | Distance Metric | | mIoU (%) |
| --- | --- | --- | --- | --- |
| | | Euclidean Norm | Cosine Similarity | |
| PN [45] | Category Prototype | ✓ | ○ | 59.9 |
| HPN [37] | | ○ | ✓ | 60.3 |
| GMM | Mixture Models | ✓ | ○ | 61.3 |
| moVMF | | ○ | ✓ | **62.4** |

## 4.3 Ablations and Analysis

All ablation studies are performed on S3DIS with PointNeXt-l as baseline.

**Distribution comparison.** We impose a comparison experiment in the distribution alignment branch of DGNet for distribution selection. The relevant experimental results are reported in Tab. 4. For category prototype models, we discard the vMF loss $\mathcal{L}_{vMF}$ and consistency loss $\mathcal{L}_{CON}$ due to the lack of corresponding forms. For the mixture model with Euclidean Norm (GMM), we replace the maximum likelihood estimation in GMM form with the $\mathcal{L}_{vMF}$ in DGNet. In terms of distance metrics, cosine similarity trumps Euclidean norm. In terms of distribution modeling, mixture models have a significant performance advantage over the category prototype models. Integrating these two aspects, the stronger fitting ability of moVMF leads to more accurate and effective supervised signals for weakly supervised learning in DGNet.

**Ablation study for loss terms.** Tab. 5 demonstrates the validity of each loss term in DGNet. Compared with partial cross-entropy loss, the truncated cross-entropy loss improves segmentation performance due to its avoidance of overfitting. Performance improvements are obtained by imposing $\mathcal{L}_{vMF}$ with soft assignment form, $\mathcal{L}_{DIS}$ and $\mathcal{L}_{CON}$ individually, and optimal performance is achieved by using these loss terms simultaneously. In contrast to the soft assignment, the hard assignment does not take into account the inter-cluster similarity and is mismatched with the soft-moVMF algorithm. Therefore, $\mathcal{L}_{vMF}$ with hard assignment form in hard-moVMF algorithm and KNN-moVMF algorithm undermines the segmentation efficiency.

**Ablation study for Nested EM Algorithm.** We ablate the proposed Nested Expectation-Maximum Algorithm in two respects. First, we optimize certain parameters on moVMF and fix other parameters with initialized values. The first, second, third, and last rows in Tab. 6 reveal that individually optimizing parts of the parameters impairs the segmentation performance. Secondly, we ablate how the parameters of moVMF are updated. Compared with kNN-moVMF (fourth raw) and hard-moVMF (fifth raw) in [3], the soft assignment strategy delivers 2.6% and 2.2% mIoU improvements, respectively. This shows that the optimization of moVMF parameters benefits from the soft assignment strategy.

**Hyperparameter selection.** In Tab. 7, we search the parameter space for suitable $\kappa$, $t$, and $\beta$. We observe that (a) the segmentation performance shows an increasing and then decreasing trend as the

Table 5: Ablation study for loss terms.

| $\mathcal{L}_{\mathrm{CE}}$ | | $\mathcal{L}_{\mathrm{vMF}}$ | | $\mathcal{L}_{\mathrm{DIS}}$ | $\mathcal{L}_{\mathrm{CON}}$ | mIoU (%) |
|---|---|---|---|---|---|---|
| $\mathcal{L}_{\mathrm{pCE}}$ | $\mathcal{L}_{\mathrm{tCE}}$ | hard | soft | | | |
| ✓ | | | | | | 58.4 |
| | ✓ | | | | | 59.1 |
| | ✓ | ✓ | | | | 58.9 |
| | ✓ | | ✓ | | | 60.4 |
| | ✓ | | | ✓ | | 59.8 |
| | ✓ | | | | ✓ | 61.1 |
| | ✓ | | ✓ | ✓ | ✓ | **62.4** |

Table 6: Ablation studies for Nested Expectation-Maximum Algorithm.

| E step | $\alpha$ | $\mu$ | mIoU (%) |
|---|---|---|---|
| None | ∘ | ∘ | 61.0 |
| soft-moVMF | ✓ | ∘ | 60.9 |
| soft-moVMF | ∘ | ✓ | 60.0 |
| hard-moVMF | ✓ | ✓ | 59.8 |
| KNN-moVMF | ✓ | ✓ | 60.2 |
| soft-moVMF | ✓ | ✓ | **62.4** |

concentration constant $\kappa$ increases. Our analysis suggests that too small $\kappa$ leads to a dispersion of features within the class, which can be easily confused with other classes. And too large $\kappa$ forces overconcentration of features within the class and overfits the network. (b) As the iteration number $t$ increases, the segmentation performance gradually rises and then stabilizes. We believe that the soft-moVMF algorithm gradually converges as $t$ increases, and increasing $t$ after convergence will no longer bring further gains to the network. (c) As the truncated threshold $\beta$ decreases, the segmentation performance shows a tendency to first increase and then decrease. The conventional cross-entropy loss function is the truncated cross-entropy loss function with $\beta = 1$. When $\beta$ decreases, the overfitting on sparse annotations is alleviated, but when $\beta$ is too small, it weakens the supervised signal on sparse labeling leading to performance degradation.

Table 7: Hyperparameter selection for the (a) concentration constant $\kappa$, (b) iteration number $t$ and (c) truncated threshold $\beta$.

| | (a) | | (b) | | (c) |
|---|---|---|---|---|---|
| $\kappa$ | mIoU (%) | $t$ | mIoU (%) | $\beta$ | mIoU (%) |
| 0.1 | 57.7 | 0 | 61.0 | 0.7 | 61.9 |
| 1 | 60.3 | 5 | 61.5 | **0.8** | **62.4** |
| **10** | **62.4** | **10** | **62.4** | 0.9 | 62.2 |
| 20 | 59.5 | 15 | 62.3 | 1 | 61.7 |

## 5 Limitations and Future Work

Despite the promising performance achieved by DGNet, exploring the distribution of embeddings is preliminary. Feng *et al.* [12] proposes a more sophisticated distribution to restrict feature learning with full supervision. However, such refinement restrictions will lead to overfitting under sparse annotations. Therefore, how to prevent weakly-supervised learning overfitting with enhanced feature description is a promising research topic.

## 6 Conclusion

In this paper, we propose a novel perspective by regulating the feature space for weakly supervised point cloud semantic segmentation and develop a distribution guidance network to verify the superiority of this perspective. Based on the investigation of the distribution of semantic embeddings, we choose moVMF to describe the intrinsic distribution. In DGNet, we alleviate the underfitting across the entire dataset and overfitting within the labeled points. Extensive experimental results demonstrate that DGNet rivals or even surpasses the recent SOTA methods on S3DIS, ScanNetV2, and SemanticKITTI. Moreover, DGNet demonstrates the interpretability of network predictions and scalability to various label rates. We expect our work to inspire the point cloud community to strengthen the inherent properties of weakly supervised learning.

## Acknowledgements

This work was supported by Shenzhen Science and Technology Program under Grant KQTD20180411143338837.

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

# Appendix A  Derivation of Optimization Objective Function

According to maximum likelihood estimation, the optimization objective function is defined as:

$$
\begin{aligned}
&\max_{\phi,\mathcal{Z},\Theta} P(\mathcal{V}|\mathcal{Z},\Theta) \\
&= \min_{\phi,\mathcal{Z},\Theta} -\prod_i P(\mathbf{v}_i|z_i,\Theta_{z_i}) \\
&= \min_{\phi,\mathcal{Z},\Theta} -\sum_i \log(\alpha_{z_i} f(\mathbf{v}_i|\kappa_{z_i},\mathbf{u}_{z_i})) \\
&= \min_{\phi,\mathcal{Z},\Theta} -\sum_i \log(\alpha_{z_i} C_d(\kappa_{z_i}) \exp(\kappa_{z_i}\mathbf{u}_{z_i}^\top \mathbf{v}_i)) \\
&= \min_{\phi,\mathcal{Z},\Theta} -\sum_i \Big[ \log(C_d(\kappa_{z_i})) + \log(\alpha_{z_i}) + \kappa_{z_i}\mathbf{u}_{z_i}^\top \mathbf{v}_i \Big].
\end{aligned}
\tag{11}
$$

To avoid the long-tail problem and simplify computations, we set a constant value for $\kappa$ on each class $c$. Therefore, the objective optimization function Eq. 11 can be further simplified as:

$$
\max_{\phi,\mathcal{Z},\Theta} P(\mathcal{V}|\mathcal{Z},\Theta) = \min_{\phi,\mathcal{Z},\Theta} -\sum_{i=1}^{n} \Big[ \log(\alpha_{z_i}) + \kappa \mathbf{u}_{z_i}^\top \mathbf{v}_i \Big].
\tag{12}
$$

---

**Algorithm 1:** soft-moVMF Algorithm

---

**Input:** Normalized Embeddings $\mathcal{V} = \{\mathbf{v}_i | i = 1, 2, \cdots, n\}$, Initial Clustering Centers $\mathcal{H} = \{\mathbf{h}_c | c \in \mathbb{C}\}$

**Output:** Soft Assignments $\mathcal{Q}$, Clustering Results $\mathcal{Z}$, Parameters of moVMF $\Theta$

/* Initialize $\alpha$, $\mathbf{u}$ in $\Theta$ */

**for** *category index $c$ of $\mathbb{C}$* **do**
     $\alpha_c, \mathbf{u}_c = \frac{1}{|\mathbb{C}|}, \mathbf{h}_c$
**end**

**repeat**
     /* The **E**xpectation step of EM */
     **for** *point index $i = 1$ to $n$* **do**
         **for** *category index $c$ of $\mathbb{C}$* **do**
             $f(\mathbf{v}_i|\kappa,\mathbf{u}_c) = C_d(\kappa)\exp(\kappa\mathbf{u}_c^\top \mathbf{v}_i)$
         **end**
         /* Compute the posterior probability $\mathbf{q}_i$ */
         **for** *category index $c$ of $\mathbb{C}$* **do**
             $P(c|\mathbf{v}_i,\Theta) = \frac{\alpha_c \exp(\kappa\mathbf{u}_c^\top \mathbf{v}_i)}{\sum_{l\in\mathbb{C}} \alpha_l \exp(\kappa\mathbf{u}_l^\top \mathbf{v}_i)}$
         **end**
     **end**
     /* The **M**aximization step of EM */
     **for** *category index $c$ of $\mathbb{C}$* **do**
         $\alpha_c = \frac{1}{n}\sum_{i=1}^{n} P(c|\mathbf{v}_i,\Theta)$
         $\mathbf{u}_c = \frac{\sum_{i=1}^{n}\mathbf{v}_i P(c|\mathbf{v}_i,\Theta)}{\|\sum_{i=1}^{n}\mathbf{v}_i P(c|\mathbf{v}_i,\Theta)\|}$
     **end**
**until** *Convergence*

**return** $\mathcal{Q} = P(\mathbb{C}|\mathcal{V},\Theta)$, $\mathcal{Z} = \underset{c\in\mathbb{C}}{\mathrm{argmax}}(\mathcal{Q})$, $\Theta = \{\alpha_c,\mathbf{u}_c | c\in\mathbb{C}\}$

---

# Appendix B  The soft-moVMF Algorithm

**Initialization.** Due to the sparsity of the annotations, some classes in the scene may lack any labeled points, thereby hindering proper initialization. Consequently, we maintain a memory bank [53] to store class prototypes across the entire dataset. Specifically, the mean embedding directions on labeled points are set as the initial vectors for categories with labeled points in the point cloud scene.

For the missing categories of this scene, we retrieve the category prototypes $\rho$ from the memory bank as supplementary initialization. Consequently, the initial vector $\mathbf{h}_c$ is formulated as:

$$\mathbf{h}_c = \begin{cases} \frac{\sum_{y_i=c} \mathbf{v}_i}{\| \sum_{y_i=c} \mathbf{v}_i \|} & c \in \mathbf{Y} \\ \rho_c & c \notin \mathbf{Y} \end{cases} . \tag{13}$$

**Pseudo Code.** As presented in Algorithm 1, we incorporate prior knowledge about the semantics during the optimization process based on soft-moVMF [3]. The weights $\alpha = 1/|\mathbb{C}|$ are initialized uniformly. Subsequently, based on the cosine similarity between features on each point and mean directions of each category, the prior probability $f$ and the posterior probability $\mathbf{q}_i$ are estimated. Finally, we determine the clustering result $z_i$ for each point by $\mathrm{argmax}(\mathbf{q}_i)$. After updating the mean directions $\mathbf{u}$ and weights $\alpha$, the process is repeated until the clustering results converge.

## Appendix C    More Experimental Results

**Impact of label rates.** To demonstrate the capability of DGNet on extreme label rates, we compare the segmentation performance on sparse annotations over a larger range of rates. Tab. 8 reports the mIoU performance of the DGNet and baseline at 10%, 1%, 0.1%, 0.01%, and 0.001% label rates. It can be observed that at 100,000 times less sparse annotations, the baseline fails to learn accurate semantic embedding from it. At the same time, our DGNet still maintains acceptable segmentation performance since it can be conducted unsupervised.

| Method | 10% | 1% | 0.1% | 0.01% | 0.001% |
|---|---|---|---|---|---|
| PointNeXt | 69.3 | 68.3 | 67.0 | 60.8 | 44.7 |
| DGNet (PointNeXt) | 69.5 | 68.8 | 67.8 | 62.4 | 51.5 |

Table 8: Performance comparison on various label rates.

**Varying labeled points.** Following SQN [18], we verified the sensitivity of DGNet (PointNeXt) to different labeled points at the same label rate. We repeated the experiment five times for each label setting, keeping the network and label rate unchanged and changing only the labeled points' locations. In Tab. 9, we observe a slight performance fluctuation within a reasonable range.

| Setting | Trail#1 | Trail#2 | Trail#3 | Trail#4 | Trail#5 | Mean | STD |
|---|---|---|---|---|---|---|---|
| 0.1% | **67.8** | 66.9 | 66.7 | 67.6 | 67.3 | 67.3 | 0.42 |
| 0.01% | 62.0 | **62.4** | 61.4 | 61.7 | 62.0 | 61.9 | 0.33 |

Table 9: Sensitivity analysis of DGNet on S3DIS Area 5.

