# OpenReview forum: "Distribution Guidance Network for Weakly Supervised Point Cloud Semantic Segmentation"
_NeurIPS.cc/2024/Conference — NeurIPS 2024 poster_

### Official Review · Reviewer_fVFW · 2024-06-28

**Soundness:** 2
**Presentation:** 2
**Contribution:** 2
**Rating:** 4
**Confidence:** 4

**Summary:**

In this work, the authors use a selected feature space distribution as supplementary supervision signals, combining with cosine similarity as the distance metric, and implement a dual-branch weakly supervised point cloud semantic segmentation network. Experiments show that using feature space distribution as supervision information can improve the performance of existing point cloud semantic segmentation networks.

**Strengths:**

This work proposes to use moVMF distribution to regulate the feature space for weakly supervised point cloud semantic segmentation. Experimental results show that this approach can improve the performances of backbone networks.

**Weaknesses:**

As the feature space distribution is selected in a fully supervised manner, it is questionable whether it is suitable for weakly supervised scenarios where labeled samples are extremely sparse. Directly comparing different distributions in the distribution alignment branch of DGNet is more convincing.

Eq. 1 in supplementary material, the last term should be log(a_zi) + k_zi * u^T_zi * vi, instead of log(a_zi) * k_zi * u^T_zi * vi.

In Eq. 9, why use the same weight for each loss term? Considering that these losses are defined from very different perspectives, their values can be of different orders of magnitude.

There are some minor flaws in the writing: some tables and figures are not referenced in the text, and some figures are not numbered in the order referenced.

**Questions:**

Please respond to my comments written above.

**Limitations:**

Evaluation on outdoor datasets is insufficient.

---

> ### Author Rebuttal · Authors · 2024-08-05
>
> # Response to Reviewer fVFW
>
> We sincerely appreciate your volunteered time. We will respond to your concerns and hopefully eliminate them.
>
> * **R1 (About comparing distributions):**
>
>   As you mentioned, we compare the fitting ability of several candidate distributions in describing the feature space from a fully supervised learning perspective in Section 3.2 of the manuscript. Based on your valuable suggestions, we impose a comparison experiment in the distribution alignment branch of DGNet for several distributions to enhance the credibility of the article. The relevant experimental results are reported in Table 3 of *rebuttal.pdf*.
>
>   We make the following modifications to embed the other three distributions individually into the distribution alignment branch of DGNet. (1) The Category Prototype models cluster features by distance magnitude, so we could not design a loss derived from maximum likelihood estimation like $\mathcal{L}\_\text{vMF}$ and the total loss used for Category Prototype models is $\mathcal{L}\_\text{tCE}+\mathcal{L}\_\text{DIS}+\mathcal{L}\_\text{CON}$. Correspondingly, there is no EM algorithm for optimizing the parameters in the distribution alignment branch. (2) For the Gaussian Mixture Model, we modify the Gaussian Mixture Model loss [1] used for images into the distribution alignment branch of DGNet.
>
>   The experimental results show that the segmentation performance of different distributions under weak supervision is consistent with the feature space fitting ability under full supervision. The stronger feature space fitting ability of moVMF under full supervision leads to more accurate and effective supervised signals for weakly supervised learning with the distribution alignment branch in DGNet.
>
> * **R2 (About typos and minor flaws):**
>
>   Thanks for the meticulous reading! The typos and minor flaws you mentioned will be rectified in the next release.
>
> * **R3 (About balance weights):**
>
>   As mentioned in line 221 of the text, instead of adjusting the weight of each loss term, DGNet significantly improves the network's performance by simply adding them together, reflecting the robust performance improvement from each loss term. Recognizing that careful tuning of the weights may lead to superior segmentation performance, we fix the $\mathcal{L}\_\text{tCE}$ and balance the weights of other loss terms by magnitude, the results of which are shown in Tab. 1 of *Rebuttal.pdf*. The balanced loss function is $\mathcal{L}\_\text{tCE}+\mathcal{L}\_\text{vMF}+0.1\mathcal{L}\_\text{DIS}+0.1\mathcal{L}\_\text{CON}$, but essentially equal in performance to the loss function used in the manuscript.
>
> * **R4 (About outdoor datasets):**
>
>   Only a few works provide segment performance on outdoor datasets in weakly supervised point-cloud semantic segmentation.  With the same baseline method RandLa-Net, DGNet improves 1.0% and 2.1% mIoU over SQN on SemanticKITTI at 0.1% label rate and 0.01% label rate, respectively. Considering the inherent density inhomogeneity of outdoor scenes, the excellent performance with such sparse annotations validates the effectiveness of DGNet with the outdoor dataset.
>
>   To enrich the evaluation of the outdoor dataset, we compare the segmentation performance of the baseline method RandLA-Net and DGNet on the *Semantic8* subset of Semantic3D. At 0.01% label rate, DGNet achieves **59.4%** mIoU compared to 55.8% of the baseline.
>
> [1] Wu, Linshan, et al. "Sparsely annotated semantic segmentation with adaptive gaussian mixtures." Proceedings of the IEEE/CVF Conference on Computer Vision and Pattern Recognition. 2023.

---

> ### Author Response · Authors · 2024-08-11
>
> Dear Reviewer fVFW:
>
> Given that the deadline for the reviewer-author discussion period is less than two days away, we are eager to discuss our work with you to clear up any concerns you may have. Overall, we have made the main efforts based on your comments as follows:
>
> * One comparison experiment in the distribution alignment branch of DGNet for several distributions.
>
> * One weight balance experiment for tuning the weights of Eq. 9.
>
> * One detailed discussion of DGNet in outdoor scenarios.
>
> We understand your time is valuable, but your opinion updates are vital to our work.

---

> > ### Comment · Reviewer_fVFW · 2024-08-12
> >
> > I keep my rating as Borderline reject, due to the insufficient evaluation on large-scale outdoor dataset, and the serious error in the objective function.

---

> > > ### Author Response · Authors · 2024-08-12
> > >
> > > Thanks for your response! We are pleased to see that three of your original five concerns have been addressed. Although you still maintain your original scores, we feel it is necessary to provide further clarification for the only two remaining.
> > >
> > > * About outdoor datasets: We sincerely disagree with your assertion that the evaluation of DGNet on outdoor scenarios is inadequate. To illustrate that our evaluation of outdoor datasets is relatively adequate in the field of weakly supervised learning, we show whether the methods are validated on these outdoor scenes by the following table:
> > >
> > >   | Method       | SemanticKITTI | Semantic3D   |
> > >   | ------------ | ------------- | ------------ |
> > >   | SQN          | $\checkmark$  | $\checkmark$ |
> > >   | OTOC         | $\times$      | $\times$     |
> > >   | CPCM         | $\times$      | $\times$     |
> > >   | PointMatch   | $\times$      | $\times$     |
> > >   | PSD          | $\times$      | $\checkmark$ |
> > >   | HybridCR     | $\checkmark$  | $\checkmark$ |
> > >   | DCL          | $\checkmark$  | $\times$     |
> > >   | MILTrans     | $\times$      | $\times$     |
> > >   | ERDA         | $\times$      | $\times$     |
> > >   | MulPro       | $\times$      | $\times$     |
> > >   | **DGNet (Ours)** | $\checkmark$  | $\checkmark$ |
> > >
> > >   This table reveals that **DGNet reaches an adequate level of outdoor validation compared with a bunch of methods.**
> > >
> > > * About typos: We guarantee that it is a transcription error without affecting the correctness of the method. **We would be regretful if our months of work were to be dismissed simply because of a typo that could be corrected immediately.**

---

### Official Review · Reviewer_ruvS · 2024-07-10

**Soundness:** 3
**Presentation:** 3
**Contribution:** 3
**Rating:** 7
**Confidence:** 5

**Summary:**

A weakly-supervised approach has been studied to relieve the annotation burden for point cloud semantic segmentation. Unlike conventional works that mainly use priors (such as similarity or augmentation-based regularization) to overcome the lack of information in weak labels, this paper introduces a novel approach directly aligning the features on the latent space of a neural network. Specifically, the proposed method employs a mixture of von Mises-Fisher distribution to model the feature distribution. This brings meaningful gains on multiple distinct architectures across various benchmarks.

**Strengths:**

1. The feature distribution is a core problem but has not yet been deeply addressed, at least in the field of weakly supervised point cloud semantic segmentation.
2. The methodology and presentation for it are pretty good.
3. The performance is impressive.

**Weaknesses:**

I think this paper has no major flaws in methodological aspects.
Therefore, I would like to ask some questions for further discussion.

1. Table 1 shows that a mixture of vMF distributions represents the distribution of normalized features better than the other choices. But why? Can you discuss the "oughtness" in more detail, at least at the level of the hypothesis? For example, why is using a mixture of vMFs better than that of Gaussians?

2. Intuitively, using a hypersphere can represent each feature as an angle, resulting in a more concise distribution. However, if the number of categories increases, this kind of approach may not effectively use the latent space, as all the points on a line map to one single point on the hypersphere (due to normalization). I wonder if the proposed method faces difficulty when handling the dataset including more categories.

3. It will be interesting if the authors check the feature distribution of the neural network trained by fully-supervised learning. Do the features still follow the mixture of vMFs well or not? Can the proposed method even enhance the performance in a fully-supervised setting?

4. Related works.
As far as I know, MPRM does not use 2D images (Line 82). And it would be better if the authors could cite recent papers in CVPR 2024.

**Questions:**

See the weaknesses.

**Limitations:**

Yes.

---

> ### Author Rebuttal · Authors · 2024-08-05
>
> # Response to Reviewer ruvS
>
> We sincerely appreciate your insightful feedback and positive evaluation. In response to your valuable comments, we structure our response as follows:
>
> * **R1 (About oughtness and moVMF):**
>
>   Next, we discuss "oughtness" and the superiority of the mixture of vMFs (moVMF).
>
>   * **Discussion about "oughtness"**.
>
>     In DGNet, the "oughtness" is a constraint on the feature space distribution under weakly supervised learning. The goal of weakly supervised learning is to enable the network to achieve comparable performance under sparse annotations as under full annotations. To achieve this, the features under weakly supervised learning should be as identical as possible to those under fully supervised learning. Due to the great discrepancy in the quality of the annotations between weakly supervised learning and fully supervised learning, obtaining the same features is difficult. Therefore, in this paper, we relax this restriction and expect to realize **the alignment of fully supervised learning and weakly supervised learning in the feature space of the neural network**. The "oughtness" means that the feature space distribution under weakly supervised learning should be identical to the feature space distribution under fully supervised learning.
>
>   *  **The superiority of moVMF**.
>
>       Above we discuss the "oughtness" of weakly supervised learning, but how to describe the feature space distribution under fully supervised learning to provide a prior for weakly supervised learning has not been addressed. Unfortunately, how to describe the feature space of a neural network is an open problem. However, it is undeniable that von Mises-Fisher has demonstrated strong data fitting and generalization capabilities in the fields of self-supervised learning [1,2], classification [3], variational inference [4], online continual learning [5], and so on.  For our explanation of the superiority of moVMF, please refer to the *R3 (About discussion and analysis of moVMF)* in the response to the Reviewer KHTz.
>
> * **R2 (About representation ability):**
>
>     Firstly, normalized features only as an input to the distribution alignment branch to provide additional supervised signals to DGNet, while the weakly supervised learning branch uses unnormalized features, and thus for the theoretical upper bound of the representation, DGNet does not differ from the baseline. Second, for the D-dimensional semantic feature $\mathbf{F}$, the normalized feature $\text{norm}(\mathbf{F})$ in the distribution alignment branch is a feature distributed on a D-1 dimensional hypersphere. Given that D is typically large (set to 256 in our implementation), the loss due to projection is almost negligible. Furthermore, we know from previous analyses that the segment head is a radial classifier, so the normalized features do not affect the segmentation results either.
>
> * **R3 (About performance on full supervision):**
>
>     The experiment in Table 1 is a validation of the neural network feature distribution under full supervision. The experimental results illustrate that the feature space distribution under full supervision fits more closely to the mixture of von Mises-Fisher distribution than the other distributions. Following your suggestion, we impose our DGNet to full supervision. With PointNeXt-L as the baseline, DGNet slightly improves mIoU from 69.2% (baseline) to **69.5%** on S3DIS, which means that distribution guidance is similarly effective for full supervision.
>
> * **R4 (About typos and reference):**
>
>     Thanks for your meticulous reading! The sentence in Line 82 should be " Via CAMs, MPRM and J2D3D dynamically generate point-wise pseudo-labels from subcloud-level annotations and image-level annotations, respectively. " We will correct this typo in the next version. We also note that some encouraging work has emerged from CVPR 2024, but these articles were not available before submission. We will cite them in the next version.
>
> [1] Chen, Xinlei, and Kaiming He. "Exploring simple siamese representation learning." Proceedings of the IEEE/CVF conference on computer vision and pattern recognition. 2021.
>
> [2] Hariprasath Govindarajan, Per Sidén, Jacob Roll, Fredrik Lindsten. "DINO as a von Mises-Fisher mixture model." ICLR 2023.
>
> [3] Scott, Tyler R., Andrew C. Gallagher, and Michael C. Mozer. "von mises-fisher loss: An exploration of embedding geometries for supervised learning." Proceedings of the IEEE/CVF International Conference on Computer Vision. 2021.
>
> [4] Taghia, Jalil, Zhanyu Ma, and Arne Leijon. "Bayesian estimation of the von-Mises Fisher mixture model with variational inference." IEEE transactions on pattern analysis and machine intelligence 36.9 (2014): 1701-1715.
>
> [5] Michel, Nicolas, et al. "Learning Representations on the Unit Sphere: Investigating Angular Gaussian and Von Mises-Fisher Distributions for Online Continual Learning." Proceedings of the AAAI Conference on Artificial Intelligence. Vol. 38. No. 13. 2024.

---

> > ### Comment · Reviewer_ruvS · 2024-08-08
> >
> > Thank you for the rebuttal.
> > Most of my concerns are addressed.
> > However, I still do not think that the explanation about the oughtness of using vMF is sufficient.
> > The main contribution of this paper is introducing vMF distribution to tackle weakly supervised learning of point cloud semantic segmentation.
> > Hence, from an academic perspective, this part should be more justified and discussed in more detail.
> > Although I would like to keep my rating, it would be better if the authors could elaborate on this further.

---

> ### Author Response · Authors · 2024-08-08
>
> Thanks for your prompt response! We are pleased that our response addresses most of your concerns. We also respect the importance you place on von Mises-Fisher distribution, and we will further elaborate on its superiority in solving the weakly supervised point cloud semantic segmentation task from multiple perspectives in our next release.
>
> We consider further analyzing the difference between the Euclidean norm (Gaussian mixture model) and cosine similarity (moVMF) from the perspective of the **Curse of Dimensionality**.
>
> * The data becomes extremely sparse in feature space as the feature dimension increases. Most feature vectors are far from each other, causing the Euclidean distance to become ineffective in distinguishing differences between feature vectors. Cosine similarity, on the other hand, is more effective in distinguishing differences between features by measuring the angle between the vectors.
>
> * Euclidean norm is very sensitive to scale. In a high-dimensional feature space, if the feature changes at different scales in different dimensions, it may result in the contribution of some dimensions to the Euclidean norm being magnified while the contribution of other dimensions is ignored. Cosine similarity is not affected by the length of the vectors, and the main measure is the angle between the features. Therefore, changes in the scale of different dimensions do not significantly affect the cosine similarity calculation.

---

### Official Review · Reviewer_KHTz · 2024-07-13

**Soundness:** 3
**Presentation:** 3
**Contribution:** 3
**Rating:** 5
**Confidence:** 4

**Summary:**

This paper addresses the problem of weakly supervised point cloud semantic segmentation. The authors propose imparting supplementary supervision signals by regulating the feature space under weak supervision. The initial investigation identifies which distributions accurately characterize the feature space in fully supervised learning, subsequently leveraging this prior knowledge to guide the alignment of weakly supervised embeddings. The authors first investigate different feature space characterizations and find that a mixture of von Mises-Fisher (moVMF) distributions with cosine similarity best describes the fully supervised embedding space. Based on this finding, DGNet is proposed with two branches: a weakly supervised learning branch and a distribution alignment branch. Leveraging reliable clustering initialization derived from the weakly supervised learning branch, the distribution alignment branch alternately updates the parameters of the moVMF and the network, ensuring alignment with the moVMF-defined feature space. Experimental results demonstrate that DGNet achieves state-of-the-art performance on multiple datasets.

**Strengths:**

1.	The proposed framework presents a novel and intriguing approach, distinguishing itself from consistency training, self-training, and similarity metric methods.

2.	The authors perform a comprehensive study to identify the most appropriate model for characterizing the feature space in fully supervised learning.


3.	DGNet is designed with two complementary branches: one dedicated to weakly supervised learning and another focused on distribution alignment. This dual-branch structure facilitates the effective integration of the distribution prior into the learning process.

4.	DGNet demonstrates its efficacy by achieving superior results across multiple datasets and various weakly supervised settings.

**Weaknesses:**

1. Computational Complexity: The paper omits a crucial discussion on the computational complexity of the distribution alignment branch, particularly regarding the EM-like algorithm. This is important, given that EM-based clustering methods traditionally tend to be computationally intensive.

2. Ablation Study: The current ablation study would benefit from exploring a broader range of loss term combinations. For instance, examining the performance of the complete loss function without $L_{DIS}$ or without $L_{CON}$ would provide more comprehensive insights into the contribution of each component.

3. Discussion and Analysis of moVMF: The paper lacks a deep analysis or discussion explaining why the mixture of von Mises-Fisher (moVMF) distribution outperforms other distributions. Additionally, it fails to provide formal guarantees on convergence or optimality. A more thorough discussion or justification for the choice of moVMF over alternative distributions would be better.

**Questions:**

See weakness

**Limitations:**

A main limitation is the computational complexity of the training.

---

> ### Author Rebuttal · Authors · 2024-08-04
>
> # Response to Reviewer KHTz
>
> We express our sincere gratitude for your valuable comments. In response to your concerns, we carefully structure our response as follows:
>
> * **R1 (About computational complexity):**
>
>   The main computational complexity of the distribution alignment branch comes from the soft-voVMF algorithm. According to the pseudocode in the Supplementary Material, the complexity of the soft-voVMF is $O(tn|\mathbb{C}|)$, where $t$ is the iteration number, $n$ is the point number of the point cloud, and $|\mathbb{C}|$ is the number of semantic categories. Since $t$, $n$, and $|\mathbb{C}|$ are all set to constant values during network training, the extra computation introduced by the distribution alignment branch is trivial. A detailed explanation of why the algorithm converges quickly at a constant $t$ is given in R3.
>
> * **R2 (About ablation study):**
>
>   Table 6 of the manuscript illustrates the ablation experiments with the loss term added individually. Based on your valuable suggestion, we show the ablation study for removing the loss terms individually in the *rebuttal.pdf*. As can be seen, each loss term contributes to the final result. Removing or modifying loss terms results in sub-optimal performance, where vMF loss has the most significant effect.
>
> * **R3 (About discussion and analysis of moVMF):**
>
>   * **Why moVMF?**
>
>     How to describe the feature space of a neural network is an open problem. Since the feature space is affected by a variety of factors such as network structure, training data, parameter settings, optimization (loss) function, *etc.*, it is unrealistic to mathematically derive an optimal and general description. Nevertheless, we construct different distributions from two dimensions and try to explain the merits and demerits of these candidate distributions in describing the feature space.
>
>     * From the distribution modeling perspective, we compare the Category Prototype Model with the Mixture Model. Despite its computational simplicity, the Category Prototype Model describes the features only by comparing the distance of features to the category prototypes. This means that the Category Prototype Model ignores the distribution within categories and the variability between categories. In contrast, the Mixture Model has a larger parameter optimization space and a relatively stronger fitting ability.
>
>     * For the distance metric, we compare the Euclidean norm and cosine similarity. We attempt to analyze these two distances with the segment head structure in the Supplementary Material. To facilitate the analysis, we simplify the structure of the segment head and define it as $\text{SegHead}(\mathbf{F})=\text{argmax}(\text{softmax}(\mathbf{WF}^\top))$, where $\mathbf{F}$ is the semantic feature extracted by the decoder and $\mathbf{W}$ is the parameter of the output layer. Consider a group of feature vectors {$k\mathbf{F} |k\geq 0 \And \mathbf{F} \neq \mathbf{0}$}. For any two feature vectors $k_1\mathbf{F}$ and $k_2\mathbf{F}$ within this group, the segmentation predictions are identical, $$\text{SegHead}(k_1\mathbf{F})=\text{argmax}(\text{softmax}(k_1\mathbf{WF}^\top))=\mathop{\text{argmax}}\limits_{i} \frac{e^{k_1 \mathbf{W}_{i,:} \mathbf{F}\_{i,:}^\top } }{\sum_j e^{k_1 \mathbf{W}\_{j,:}\mathbf{F}\_{j,:}^\top}} = \text{argmax}(\text{softmax}(\mathbf{WF}^\top))=\text{argmax}(\text{softmax}(k_2\mathbf{WF}^\top))=\text{SegHead}(k_2\mathbf{F}).$$ If the general case of using an activation function is taken into account, it does not change the result after argmax since the activation function is usually monotonically nondecreasing. Therefore, **the segment head is a radial classifier** that has a more pronounced classification performance on the angles, so that cosine similarity describes the feature space better than the Euclidean norm.
>
>     In summary, the mixture of von Mises-Fisher distribution with both cosine similarity and mixture model is optimal among the candidate distributions, and the experimental comparisons in Tab 1 validate our analysis.
>
>   * **Convergence proof.**
>
>     The soft-moVMF belongs to the EM algorithm. In 1983, Jeff Wu proved the convergence properties of the EM algorithm in [1]. He proved that the EM algorithm can make the log-likelihood function of the observed data converge to a stable value under certain conditions.
>
>     However, this stable value is not necessarily optimal, which is related to the initialization of the EM algorithm. Therefore, we prove that the difference between the initialization $\mathbf{h_c}$ from the weakly supervised learning branch and the theoretical optimal mean vector $\mathbf{u^*\_c}$ conforms to a Gaussian distribution with a mean value of 0. According to maximum likelihood estimation, $\mathbf{u}^*\_c=\frac{1}{n_c}\mathop{\sum}\limits_{y\_i = c} (\mathbf{F}\_i)$, where $n_c$ denotes the point number belonging to c-th category and $y_i$ denotes the i-th label of full annotations. The initialization $\mathbf{h_c}$ from the weakly supervised learning branch can be defined as $\mathbf{h_c}=\frac{1}{m_c}\mathop{\sum}\limits_{y'_i = c} (\mathbf{F}_i)$, where $m_c$ is the labeled point number belonging to c-th category in sparse annotations and $y'_i$ is the i-th label of sparse annotations. Sparse labeling can be considered as the result of sampling over the full labeling, *i.e.*, $y'=s(y)$. Assume that the sample is independently redistributed and obeys a certain distribution $\mathcal{D}(\mu,\sigma^2)$. According to the Central Limit Theorem, the following convergent distribution can be obtained: $$(\mathbf{u}^*\_c-\mathbf{h}\_c) \sim \mathcal{N}(0, \frac{\sigma^2}{m_c}).$$ This means that the initialized mean vector in DGNet has a high probability of appearing in the vicinity of the optimal solution. Therefore, the soft-moVMF converges quickly at a constant iteration number $t$.
>
>  [1] Wu, CF Jeff. "On the convergence properties of the EM algorithm." The Annals of statistics (1983): 95-103.

---

> > ### Author Response · Authors · 2024-08-11
> > **The analysis of moVMF from the perspective of the Curse of Dimensionality**
> >
> > Dear Review KHTz:
> >
> > The discussion and analysis of moVMF are further enriched during the conversation with the Review ruvS. Specifically, we consider further analyzing the difference between the Euclidean norm (Gaussian mixture model) and cosine similarity (moVMF) from the perspective of the **Curse of Dimensionality**.
> >
> > * The data becomes extremely sparse in feature space as the feature dimension increases. Most feature vectors are far from each other, causing the Euclidean distance to become ineffective in distinguishing differences between feature vectors. Cosine similarity, on the other hand, is more effective in distinguishing differences between features by measuring the angle between the vectors.
> >
> > * Euclidean norm is very sensitive to scale. In a high-dimensional feature space, if the feature changes at different scales in different dimensions, it may result in the contribution of some dimensions to the Euclidean norm being magnified while the contribution of other dimensions is ignored. Cosine similarity is not affected by the length of the vectors, and the main measure is the angle between the features. Therefore, changes in the scale of different dimensions do not significantly affect the cosine similarity calculation.

---

> ### Author Response · Authors · 2024-08-11
>
> Dear Reviewer KHTz:
>
> Given that the deadline for the reviewer-author discussion period is less than two days away, we are eager to discuss our work with you to clear up any concerns you may have. Overall, we have made the main efforts based on your comments as follows:
>
> * A crucial discussion on the computational complexity.
>
> * An ablation study for removing the loss terms individually.
>
> * The discussion and analysis of the superiority of moVMF.
>
> We understand your time is valuable, but your opinion updates are vital to our work.

---

### Official Review · Reviewer_bWaK · 2024-07-13

**Soundness:** 2
**Presentation:** 2
**Contribution:** 2
**Rating:** 4
**Confidence:** 5

**Summary:**

This paper outlines a study focusing on distance metric and distribution modeling, highlighting the effectiveness of combining mixture of von Mises-Fisher distributions (moVMF) with cosine similarity. A Distribution Guidance Network (DGNet) is introduced, featuring two main branches: weakly supervised learning and distribution alignment. The network utilizes reliable clustering initialization from the weakly supervised branch to iteratively update moVMF parameters and network parameters, ensuring alignment with moVMF-defined feature spaces.

**Strengths:**

Originality: DGNet is a distribution guidance network that combines a weakly supervised learning branch and a distribution alignment branch. The weakly supervised learning branch makes use of reliable cluster initialization for learning. This means that the network is able to perform preliminary learning and classification of data through clustering methods without detailed labeling. This approach helps to reduce the dependence on large amounts of labeled data and improves the robustness and scalability of the model. The distribution alignment branch alternately updates the parameters of the moVMF (von Mises-Fisher distribution) and the parameters of the network to ensure that the network parameters are aligned with the feature space defined by the moVMF. This method helps to guide the network to learn the structure and characteristics of data distribution, so as to improve the generalization ability and adaptability of the model.

Quality: DGNet explains the prediction probability from the perspective of Bayes' theorem and provides the confidence of each prediction, which proves the rationality of DGNet selection distribution and network design. In addition, extensive experiments confirm the soundness and efficacy of the chosen distribution and network design.

**Weaknesses:**

1. The section of 4.2 and 4.3 can be written in more detail. In particular, the explanation of changes in model performance in ablation experiments should be added. Specifically, a thorough explanation of the changes in model performance observed during ablation experiments should be incorporated. Ablation experiments involve systematically removing components or features from the model to analyze their impact on performance. By detailing how each modification affects the overall performance metrics, readers gain insights into the critical elements contributing to the model's effectiveness.
2. In Abstract, there is no a clear problem to address from existing methods of weakly supervised learning for 3D point clouds. Similarly, in Introduction, it’s also not clear about the motivation of this work and existing problems of this topic.
3. There is no main contributions about this work in Introduction.
4. How to balance weights of each loss function in Equation (9)?
5. There is no any qualitative comparison between this work with others.
6. Some quantitative results from this work are also incremental improvements compared to others within one percentage point.
7. Compared with previous work, the performance improvement of DGNet model may not be substantial enough to justify its superiority over existing methods, especially on the ScanNet and SemanticKITTI datasets.
8. In the section of 4.3, why does the performance of the model degrade when optimizing some parameters? What is the impact of parameter optimization on resource consumption?

**Questions:**

Please answer the questions in Weaknesses.

**Limitations:**

The authors have thoroughly discussed the limitations of DGNet, and there are no potential negative societal impacts associated with their work.

---

> ### Author Rebuttal · Authors · 2024-08-02
>
> # Response to Reviewer bWaK
> We appreciate your valuable comments and suggestions. We will respond to each of your concerns and hopefully eliminate them.
>
> * **R1 (About detailed explanation):**
>
>   Due to page constraints, we analyze and interpret the main and important experimental results in the manuscript. We will provide a detailed elaboration of the analysis of the experimental results in Sec. 4.2 and 4.3 during the reviewer-author discussion period, due to the character limit in the rebuttal period.
>
> * **R2 (About writing):**
>
>   In our writing experience, there are generally three main types of writing in academic articles:
>   * The article proposes a new task, which requires a description of the significance of the new task and the difficulties in solving it.
>   * The article provides specific improvements to the existing methodology, which needs to describe the problems faced by the methodology.
>   * The article develops a new paradigm to address the current challenges, which entails summarizing the previous paradigms and elaborating on the advantages of the proposed new paradigm.
>
>   We believe that DGNet is more per the third writing logic. Therefore, we focus on the challenges of weakly supervised point cloud semantic segmentation in the Abstract. In the second paragraph of the Introduction, we summarize current methods into three paradigms and draw out their common problem, for which we propose a new paradigm.
>
> * **R3 (About contribution summary):**
>
>   According to our understanding, NeurIPS 2024 does not have a mandatory requirement to summarize contributions. Some highly cited papers also do not summarize major contributions, such as ResNet, Attention is All You Need, Non-local Neural Networks, ViT, *etc*.
>
> * **R4 (About balance weights):**
>
>   As mentioned in line 221 of the text, instead of adjusting the weight of each loss term, DGNet significantly improves the network's performance by simply adding them together, reflecting the robust performance improvement from each loss term. Recognizing that careful tuning of the weights may lead to superior segmentation performance, we fix the $\mathcal{L}\_\text{tCE}$ and balance the weights of other loss terms by magnitude, the results of which are shown in Tab. 1 of *Rebuttal.pdf*. The balanced loss function is $\mathcal{L}\_\text{tCE}+\mathcal{L}\_\text{vMF}+0.1\mathcal{L}\_\text{DIS}+0.1\mathcal{L}\_\text{CON}$, but essentially equal in performance to the loss function used in the manuscript.
>
> * **R5 (About qualitative comparison):**
>
>   Our work follows the visualization experimental setup of state-of-the-art methods (such as ERDA, CPCM, PointMatch, *etc*) in the field of weakly supervised point cloud semantic segmentation, focusing on visual comparisons between the baseline and the proposed method.
>
> * **R6 & R7 (About performance improvement):**
>
>   For a detailed explanation of performance, including why DGNet has less boost in some cases, please refer to the specific response in R1.
>
>   We would like to emphasize that DGNet is not an incremental work, *i.e.*, it is not an adaptation or refinement of an existing weakly supervised learning paradigm, but rather a new weakly supervised learning paradigm constructed from the perspective of describing the distribution of feature space. Therefore, the purpose of the comparison experiments is to demonstrate that this novel weakly supervised learning paradigm is feasible and has the potential to outperform other weakly supervised learning paradigms. We recognize that there is still room for performance improvement in some datasets, but we believe the current comprehensive performance benefits across multiple datasets and multiple sparse labeling settings are sufficient to demonstrate the potential of this new paradigm. We look forward to the subsequent ongoing efforts of the Point Cloud community to fully exploit the weakly supervised performance of this paradigm.
>
> * **R8 (About performance degradation and resource consumption):**
>
>   * **Explanation for performance degradation** We find two places that could be considered performance degradation in Sec. 4.3. The first place is the introduction of $\mathcal{L}\_\text{vMF}$ with hard assignment form. In contrast to the soft assignment, the hard assignment does not take into account the inter-cluster similarity and is mismatched with the soft-moVMF algorithm. So DGNet uses $\mathcal{L}\_\text{vMF}$ with soft assignment form. The second place is to optimize $\alpha$ and $\mathbf{u}$ individually because the fixed parameters affect the accurate update of the optimizable parameters. Therefore, DGNet simultaneously optimizes $\alpha$ and $\mathbf{u}$.
>   * **Explanation for resource consumption** In the inference phase, there is no additional memory consumption compared to the baseline since DGNet only activates the weakly supervised learning branch for inference. In the training phase, the resource consumption mainly comes from the Memory Bank and Soft-moVMF algorithm. Compared to the baseline, DGNet only increases 25.8M during the training phase, and this resource consumption increase is not significant compared to the baseline network itself.

---

> ### Author Response · Authors · 2024-08-08
> **A detailed explanation for R1**
>
> Next, we provide a detailed elaboration of the analysis of the experimental results in Sec. 4.2 and 4.3. **Please note that these additions of details do not detract from the main conclusions in the paper, and we believe that some details are so burdensome that may dilute the focus on key experiments and make the paper lengthy.**
>
> * **Results on S3DIS.** The gain that DGNet brings to the baseline method is more pronounced at 0.01% label rate compared to 0.1% label rate, since the guidance on feature distribution is more valuable with extremely sparse annotations. The 0.02% label rate denotes a sparse labeling form of "one-thing-one-click", and unlike methods that use this labeling form, we did not introduce super-voxel information to enrich the original sparse labeling. However, DGNet outperforms these methods at a smaller label rate. In addition, we would like to re-emphasize that DGNet demonstrates strong generalization capabilities, both in terms of different annotation scales and various baseline methods.
>
> * **Results on ScanNetV2.** Confronted with the diverse categories and versatile scenes of ScanNetV2, we follow the other methods and report on the segmentation performance with the sparse annotations processed by super-voxels. The cross-entropy loss term can generate relatively sufficient supervised information to train the network due to the introduction of pseudo-labelings, resulting in a less pronounced improvement of DGNet than that of S3DIS. However, DGNet is still slightly superior to the latest SOTA methods.
>
> * **Results on SemanticKITTI.** In the field of weakly supervised point-cloud semantic segmentation, only a few works provide segment performance on outdoor SemanticKITTI.  With the same baseline method RandLa-Net, DGNet improves 1.0% and 2.1% mIoU over SQN at 0.1% label rate and 0.01% label rate, respectively. Considering the inherent density inhomogeneity of outdoor scenes, the excellent performance with such sparse annotations validates the effectiveness of DGNet with the outdoor dataset.
>
> * **Interpretability of prediction results.** The interpretation of Bayesian posterior probabilities for trained networks based on the distribution function of the moVMF is an attractive advantage of DGNet. Figure 3 visualizes the posterior probabilities for some categories. Taking the “floor” as an example, according to the Bayesian theorem, those points with relatively high posterior probabilities have high probabilities of belonging to the floor, which is also consistent with the network predictions.
>
> * **Hyperparameter selection.** In Table 5, we search the parameter space for suitable $\kappa$, $t$, and $\beta$. We observe that (1) the segmentation performance shows an increasing and then decreasing trend as the concentration constant $\kappa$ increases. Our analysis suggests that too small $\kappa$ leads to a dispersion of features within the class, which can be easily confused with other classes. And too large $\kappa$ forces overconcentration of features within the class and overfits the network. (2) As the iteration number $t$ increases, the segmentation performance gradually rises and then stabilizes. We believe that the soft-moVMF algorithm gradually converges as $t$ increases, and increasing $t$ after convergence will no longer bring further gains to the network. (3) As the truncated threshold $\beta$ decreases, the segmentation performance shows a tendency to first increase and then decrease. The conventional cross-entropy loss function is the truncated cross-entropy loss function with $\beta=1$. When $\beta$  decreases, the overfitting on sparse annotations is alleviated, but when $\beta$  is too small, it weakens the supervised signal on sparse labeling leading to performance degradation.
>
> * **Ablation study for loss terms.** Table 6 demonstrates the validity of each loss term in DGNet. It is worth noting that $\mathcal{L}\_\text{vMF}$ with hard assignment form undermines the segmentation efficiency, on the contrary, $\mathcal{L}\_\text{vMF}$ with soft assignment form has a clear positive effect. In contrast to the soft assignment, the hard assignment does not take into account the inter-cluster similarity and is mismatched with the soft-moVMF algorithm. In addition, according to suggestions from another reviewer, we supplement this ablation experiment with more ablative forms. Please refer to the response to the Reviewer KHTz.
>
> * **Ablation study for soft-moVMF.**  The experimental results in the first, second, third, and last rows of Table 7 show that it is optimal to optimize both $\alpha$ and $\mathbf{u}$ with the soft-moVMF algorithm. Optimizing $\alpha$ and $\mathbf{u}$ individually impairs the segmentation performance, for which the fixed parameters affect the accurate update of the optimizable parameters. The forth, fifth and last rows of Table 7 show that the soft-moVMF outperforms hard-moVMF and KNN-moVMF due to more accurate parameter updates.

---

> ### Author Response · Authors · 2024-08-11
>
> Dear Reviewer bWaK:
>
> Given that the deadline for the reviewer-author discussion period is less than two days away, we are eager to discuss our work with you to clear up any concerns you may have. Overall, we have made the main efforts based on your comments as follows:
>
> * Further explanation of the experimental results.
>
> * One weight balance experiment for tuning the weights of Eq. 9.
>
> * Detailed clarification of the article writing.
>
> We understand your time is valuable, but your opinion updates are vital to our work.

---

> > ### Comment · Reviewer_bWaK · 2024-08-12
> >
> > After reading the authors' responses and comments from other reviewers, I do not change my final score due to the following:
> > 1. As shown in Table 1, compared with the Category Prototype (CP) model plus cosine similarity, the proposed method only shows an incremental improvement. And the method does not have any theoretical certificates.
> > 2. Further, the comparison is insufficient because the authors do not directly compare their method with CP and other distributions under different weak label ratios.
> > 3. Regarding the Qualitative comparison, this paper does not give insufficient results (only a few in the manuscript), and they also can not be found in the Appendix.
> > 4. Some Symbols do not have definitions and are even wrong, e.g., there is no Kc in Equation (5).

---

> > > ### Author Response · Authors · 2024-08-12
> > >
> > > Thanks for your response. We believe that your final comments are basically from other reviewers and have already been responded to by us. Even though you did not change your final score, we would still like to clarify these points of concern you listed as a matter of courtesy and respect:
> > >
> > > * About improvement and certificates. The category prototype model with cosine similarity and the final adopted moVMF are compared under our proposed distributional alignment framework. **There is a logical problem with taking two different distributional choices under the same framework to show that the framework is not valid.** In addition, we do not agree that there is no theoretical proof for the method you claim. We devote considerable space throughout the rebuttal phase to theorizing the superiority of moVMF, please refer to R3 to reviewer KHTz and R1 to reviewer ruvS.
> > >
> > > * About comparison. We believe that your concern stems from reviewer fVFW's question 1. However, this concern has been eliminated based on our experiments and the response of reviewer fVFW. Please refer to R1 to reviewer fVFW.
> > >
> > > * About qualitative comparison. We think that some representative samples of visualization comparisons are sufficient to demonstrate the validity of the methodology under the current page limitations. What additional conclusions would you like to draw from more qualitative results?
> > >
> > > * About typos: We guarantee that it is a transcription error without affecting the correctness of the method. **We would be regretful if our months of work were to be dismissed simply because of a typo that could be corrected immediately.** Moreover, the $\kappa_c$ is ignored in Eq. 5 because the $\kappa_c$ is a constant value for different catergories.

---

### Author Rebuttal · Authors · 2024-08-06

# Summary of Author Rebuttal

We respectfully appreciate the constructive comments and valuable suggestions given by each reviewer. We are confident that each reviewer has given sufficient time to scrutinize our work. To respond to the issues raised and to dispel some misconceptions, we analyze and explain each comment. All the effects we made in the rebuttal period can be summarized in three aspects:

* **Explanation of distribution choice.** We re-emphasize the superiority of the chosen mixture of the von Mises-Fisher distribution (moVMF) from several perspectives, including the distribution modeling, the structure of the segment head, and the feature-fitting ability. We also expect that this work will stimulate further exploration in the point cloud community concerning the feature space description.

* **Supplementation of experiments.** Based on the suggestions, our main supplementary experiments include (1) a weight-balancing experiment for loss terms, (2) an ablation experiment for removing loss terms, and (3) a comparison experiment for different distribution priors in the distribution alignment branch. Due to the character limitations of the dialog box, we place the results of the three experiments in *rebuttal.pdf*.

* **More interpretation for DGNet.** We explain the writing logic of the article, the complexity of the algorithm, the convergence of the algorithm, the representation ability of the features, and so on. These explanations provide a more consolidated theoretical support for DGNet.

We expect that these responses during the rebuttal period will address the main concerns raised by reviewers, and we also look forward to further polishing this work with the experts during the reviewer-author discussion period.

---

### Decision · Program_Chairs · 2024-09-25

**Decision:**

Accept (poster)

**Comment:**

This paper receives bipolar ratings of 1 accept, 1 borderline accept, and 2 borderline reject. Most reviewers agree the core problem of investigating feature distribution raised by the paper is important and interesting, and the authors did sufficient experiments on the benchmark datasets. However, there are few concerns raised regarding the oughtness of the selected moVMF distributions, incremental improvements against existing methods, and lack of evaluation on large-scale outdoor datasets. Some reviewers' concerns are addressed by the rebuttal but not all of them.

After carefully reading the paper, reviews, and the rebuttal discussion, AC finds that the paper is well-written, logically sound with adequate evaluation. Therefore, AC thinks the paper is ready and suitable for publication in NeurIPS. The authors are encouraged to add more justification on the oughtness of moVMF, given cosine similarity simply outperforms euclidean distance in high dimensional space and moVMF is the only distribution candidate in the design space. We congratulate the authors on the acceptance of their paper!